# **Muon paleotopometry**

John C. Gosse<sup>1</sup>, Alan J. Hidy<sup>2</sup>, Lauren MacLellan<sup>1</sup>, Joel Pederson<sup>3</sup>, Maya Soukup<sup>1</sup>, Gerald Raab<sup>1</sup>, Matt Drew<sup>1,4</sup>, Sophie Norris<sup>1,5</sup>, Marie-Cécile Piro<sup>6</sup>, and William U. Woodley<sup>6</sup>

- <sup>1</sup>Department of Earth and Environmental Sciences, Dalhousie University, Halifax, NS, Canada
  - <sup>2</sup>Center of Accelerator Mass Spectrometry, Lawrence Livermore National Laboratory, Livermore, CA, USA
  - <sup>3</sup>Department of Geosciences, Colorado State University, Fort Collins, CO, USA
  - <sup>4</sup>Department of Earth Sciences, Memorial University, St. John's, NL, Canada
  - <sup>5</sup>Department of Geography, University of Victoria, Victoria, BC, Canada
- 10 <sup>6</sup>Department of Physics, University of Alberta, Edmonton, Canada

Correspondence to: John C. Gosse (John.Gosse@Dal.Ca)

Abstract. Recent advances in measuring muon fluxes at great depths for neutrino experiments, along with improvements in in-situ terrestrial cosmogenic nuclide (TCN) measurements, have enhanced our understanding of muogenic nuclide production rates at decametre and hectometre depths. These developments allow us to explore muogenic TCN as a tool for estimating long-term (>10<sup>5</sup> years) erosion rates. There are theoretical advantages of utilizing muogenic TCN for long-term landscape evolution analysis (μ-paleotopometry). We summarize recent advances in knowledge of deep muon flux used in the interpretation of neutrino interactions at kilometre depths. We discuss strategies being considered for the μ-paleotopometry method to address otherwise intractable landscape evolution questions. We demonstrate an achievable resolution of calculated erosion rates with TCN measurements in quartz at rock depths of 37.8 x 10<sup>3</sup> to 67.5 x 10<sup>3</sup> g cm<sup>-2</sup> (~140 and 250 m). In settings where assumptions of constant erosion rate are suitable, the uncertainty is controlled by nuclide concentration measurement error and the effective time (duration over which the isotope concentration reflects the erosion history) is limited by radio-decay. Other environmental complexities, such as variable glacier cover, unknown complexity in initial topography, or mineral composition of rock may restrict the addressable questions and limit precision. Where a time-varying erosion rate is sought, deeper-time variances have inferior representation.

#### 1 Introduction

A challenge for understanding landscape evolution over 10<sup>5</sup> and 10<sup>6</sup> yr timeframes is quantifying rates of denudation and incision. Low temperature thermochronometers (e.g. apatite (U+Th)/He) provide thermal histories which can be interpreted as exhumation rates, but for the Quaternary, most thermochronometers are applicable in tectonically active orogens or high relief regions with sufficient weathering and transport rates (Herman et al., 2010). Quaternary valley incision has been quantified by measuring heights and ages of stream straths or estimating them by dating gravels over the strath or related trapped (e.g. cave) sediments. Unfortunately, the probability of preservation of terraces and other gravels diminishes with time and caves are uncommon in non-karst settings. Stream vertical position can also be directly dated with TCN to establish incision rates by dating continuously exposed and non-eroded stream-polished surfaces on valley wall cliffs which have been exposed longer than the streams aggradation-incision cycles; unfortunately, such sequences are rare. Catchment-wide average erosion rate approaches using terrestrial cosmogenic nuclides (TCN) extracted from paleo-stream sediments can provide average rates of erosion on the catchment slopes for a particular duration (typically past  $10^3 - 10^4$  yr depending on erosion rate and isotope (Hidy et al., 2014)), but may not be applicable for complex (e.g. landslide-prone) landscapes or glaciated regions where concentrations in regolith have not achieved steadiness. Nevertheless by measuring the inherited concentrations in eroded catchment sediments deposited in datable terraces or basin strata it is possible to obtain catchment erosion histories spanning beyond 10<sup>5</sup> years, but reworking of buried sediment may be an undefinable source of error (Sinclair et al., 2018). Incision through a series of basalts where the age of the uppermost lava is known is a rare example where amount and duration of

erosion can be constrained (Mitchell et al., 2023). An applied challenge posed by nuclear waste management agencies is quantifying erosion rates at a specific location—typically not in a river valley—over a regulatory period of one million years in low relief regions with relatively slow erosion rates. A new strategy is needed to bridge the limits of thermochronometry and chronometry of surface features.

Landscape erosion rate can be quantified if the amount of erosion (mass (g), depth (cm), or mass depth (g cm<sup>-2</sup>)) and the duration of erosion ( $T_e$  (Myr)) are known. While the amount of erosion is sometimes discernible, e.g. volume of an incised valley or delta, the time that erosion began throughout the system is usually not. Muon-paleotopometry offers a means to quantify bedrock erosion history of a location on Earth and relief generation over million-year timescales, by exploiting the predictable depth dependence of the concentration of muogenic terrestrial cosmogenic nuclides ( $\mu$ TCN). The method quantifies the erosion-induced change in shielding (i.e. difference in initial and final shielding thicknesses,  $\Delta$ S = S<sub>i</sub> – S<sub>f</sub>) above a subsurface sample. It can be applied in a relative sense, such as the creation of a valley relative to an interfluvial plateau (relief generation), or in a direct sense for the long-term (e.g.  $10^5$ - $10^7$  yr) erosion rate at a specific location (Fig. 1).

Figure 1. Elements of  $\mu$ -paleotopometry. The reduction in shielding ( $\Delta S$ ) above a sample owing to net erosion is the difference between the initial shielding thickness ( $S_i$ ) and the final shielding thickness ( $S_f$ ).  $\Delta S$  is the amount of material eroded from the paleosurface. R shows the location of maximum relief in the cross section. Orange and blue circles represent vertical and horizontal strategies for sampling. Pink shadings around an arbitrary sample are a reminder that the cosmic ray secondary flux to any sample will depend on topography and density variation in the crust and may originate slightly below horizontal. Approximately 99% of the muon flux is sourced within a 75° cone about zenith.

Here we summarize two approaches to  $\mu$ -paleotopometry in erosional landscapes, provide a proof of concept to establish that measurements of some cosmogenic nuclides at depths of hectometres are achievable at required precisions, and discuss sources of uncertainty and limitations to the methods. We indicate pertinent knowledge and software now available to compute muon flux at these depths, and how  $\mu$ -paleotopometry may be applied for samples deeply in crust with varying topography and

density. We learn that, in certain circumstances, using multiple nuclides and a premeditated sampling strategy it is possible to estimate duration and variable rates of thinning or thickening in the upper crust.

# 2 Background

Primary galactic cosmic radiation (GCR), mostly protons, have sufficiently high energy to penetrate Earth's magnetic field and interact with nuclei of atoms in the atmosphere to produce a cascade of different secondary radiation which continues to react with atmospheric, hydrospheric, and lithospheric nuclei, diminishing in average flux energy with depth. Secondary cosmogenic nuclides and other particles are produced by numerous interactions. In rocks at near Earth's surface, TCN such as <sup>3</sup>He, <sup>10</sup>Be, <sup>14</sup>C, <sup>21</sup>Ne, <sup>26</sup>Al, <sup>36</sup>Cl, and <sup>41</sup>Ca are produced mostly from spallation interactions with fast nucleons (n<sub>f</sub>); muogenic interactions (see below) account for about 1-3% of <sup>10</sup>Be and <sup>26</sup>Al surface production, more for others (e.g. <sup>14</sup>C), and thermal and epithermal capture interactions are important for only a few TCN such as <sup>36</sup>Cl and <sup>41</sup>Ca (Gosse and Phillips, 2001; Borchers et al., 2016). The flux of fast nucleons diminishes more rapidly with depth (attenuation length,  $\Lambda_{nf}$ , approximately 135-160 g cm<sup>-2</sup> from low to high latitude) than muons ( $\Lambda_{\mu} \ge 1500$  g cm<sup>-2</sup>) mainly because muons are smaller (nine times less massive). The kinetic energy of secondary muons spans a spectrum: as muons traverse matter, those with high-energy (referred to as "fast" muons ( $\mu_f$ ) in the TCN and other fields) lose energy through ionization, bremsstrahlung, pair production, and photonuclear interactions, gradually evolving into lower-energy ("slow") muons (u<sub>s</sub>) which eventually stop or decay. In the lithosphere and hydrosphere, most muons stop, capture electrons, form muonium, or undergo deepinelastic scattering with rock nuclei accompanied by neutron emission (Fedynitch et al., 2022). Low-energy ("slow") muons contribute to TCN production  $[\sigma_{us \to TCN} \sim mb]$  mostly via muon capture, a process where a negative muon is attracted to an atomic nucleus and interacts with a proton to produce a neutron and muon neutrino (Heisinger et al., 2002a). In contrast, high-energy  $\mu_f$  (i.e. >10 GeV according to GEANT-4, a Monte Carlo program for simulating the geometry and tracking of the particle flux through matter (Collaboration and Agostinelli, 2003)) can produce TCNs through direct muon-induced photonuclear interactions, where a virtual photon mediates energy transfer, leading to inelastic nuclear scattering analogous to neutron-induced spallation  $[\sigma_{uf} \rightarrow_{TCN} \sim \mu b, (Heisinger et al., 2002b)]$ . Furthermore, high-energy muons can emit real photons via bremsstrahlung, which may trigger secondary photonuclear reactions—providing an additional, albeit typically minor, pathway for TCN production. Some of these interactions also generate secondary particles, particularly neutrons, with sufficient energy to induce spallation reactions in surrounding material. Other high-energy processes, such as deep inelastic scattering are theoretically possible but are generally rare at typical underground energies.

At depths greater than decametres, TCN production is dominated by  $\mu_f$ . While there remains some uncertainty in the energy spectra for  $\mu_f$  over million-year timescales, and in the nuclear cross sections for some interactions at those energies, knowledge of muon flux and energies from decades of theoretical and experimental physics is converging with lessons from the cosmogenic isotope community, particularly new knowledge from the dark matter community which requires knowledge of




muon flux at deep underground laboratories. For example, experimental measurements of the attenuation through cement for slow (Heisinger et al., 2002a) and fast (Heisinger et al., 2002b) muons with known discrete energies and angular incidence distribution have been reasonably reproduced by measurements of cosmogenic  $^{10}$ Be and  $^{26}$ Al in a very slowly eroding quartzrich rock in Antarctica at depths to 6500 g cm² below surface (Balco, 2017). Another example of convergence is the flux and energy spectra of fast muons at depths below 1000 g cm², normalized for a range of angular incidence distributions at different elevations, depths, and topography, as computed by programs such as GEANT-4 with MUTE (MUon intensiTy codE) (Fedynitch et al., 2022; Woodley et al., 2024) have very closely reproduced the  $\mu$ -flux measured with muon detectors at deep subsurface laboratories, such as the  $2092 \pm 6$  m deep Sudbury Neutrino Observatory Lab (SNOLab, (Aharmim et al., 2009); Fig. 2). The relationship and sensitivity of  $\mu$ TCN concentrations to erosion rate have been recognized previously (Kim and Englert, 2004; Balco, 2017), but the uncertainties in relevant key physical parameters and the imprecision of measurement of  $\mu$ TCN at deep depths remain challenges.

Figure 2. Comparison of the computed (MUTE) (Fedynitch et al., 2022) and measured (Aharmim et al., 2009) total underground muon fluxes at different laboratories (Woodley et al., 2024). The MUTE (v. 2.0) curve (solid black) is fit to the flux for each lab (blue circles) assuming a flat overburden. The coloured symbols represent the most recent experimental flux values at each site. The coloured symbols with thick black outlines are calculated with MUTE using topographic maps and rock compositions for the overburden.

We can obtain a mean constant erosion rate ( $\varepsilon$ ) of continuously exposed surfaces (Lal, 1991). To do this we assume that the surface has been steadily eroding for sufficient time that its concentration is constant (steady) and only controlled by erosion rate, such that

$$\varepsilon \sim \frac{P\Lambda}{N\rho}$$

where P is the time-averaged production rate of the TCN in atom  $g^{-1}$  yr<sup>-1</sup> over the exposure duration,  $\Lambda$  is the attenuation length (g cm<sup>-2</sup>) for the incident cosmogenic secondary particles that produce the TCN,  $\rho$  is bulk density (g cm<sup>-3</sup>) of the eroding rock,







and N is the measured TCN concentration in the surface minerals (atom g<sup>-1</sup>) (simplified from Lal 1991 eqns. 8 and 11). The difference between the average erosion rate of the samples from a plateau and a valley incised through it is an average rate of relief-production. Lal's approach uses surface samples and single or multiple nuclides, and the concept can be modified to consider many production pathways for each TCN, decay if the TCN is radioactive, time-varying production rates and erosion rates, and measuring the shape of the TCN concentration vs. depth profile (e.g. exponential for fast nucleons) in the upper few metres of rock. This can be extended deeper underground to use muogenic production of one or more isotopes. The shape of the concentration vs. depth curve will provide a unique erosion rate if we use a noble gas (stable) isotope (<sup>3</sup>He, <sup>21</sup>Ne), or the duration of continuous exposure exceeds the time required for secular equilibrium ('steady-state') of the concentration of a TCN. At higher erosion rates, radionuclides will require less time to achieve steady state concentration, and that duration is even shorter for shorter-lived radioisotopes. Thus, there are advantages to using multiple nuclides with different decay rates to capture erosion rates for different durations. The ideal situation for the application of any TCN method for calculating erosion rate for a particular location is a simple surface (i.e. over the duration of exposure, there are changes in no overburden cover, the horizontal surface above a sample remains horizontal, and that all other parameters such as density or production rates spatially and temporally constants or known) and the rate of erosion is constant relative to the method used. These attributes enable the assumption of steadiness of the concentration with depth. The duration needed for predictable concentration steadiness is a function of erosion rate and—in the case of radionuclides—decay rate, so shorter lived radionuclides may be useful to allow steadiness to be assumed over more recent exposure periods (e.g. post-deglaciation, or post-acceleration of erosion).

In addition to the predicable concentration dependence on mass depth and erosion rate, there is a geometrical feature in the concentration vs depth curves that is also dependent on erosion rate. As the spallation and slow-muon production rate curves are both approximately exponential with depth, their intersection occurs where the (more penetrative) muons create relative more concentration as the flux of nucleons diminishes with depth. This creates an inflection, or knee, in the summed concentration with depth profile, at approximately  $10^3$  g cm<sup>-2</sup> (i.e.  $\sim 3.7$  m, with  $\rho_{\text{bulk}} = 2.68$  g cm<sup>-3</sup>) depth below surface (Fig. 3). This knee has been measured in nature (cf. (Balco, 2017; Kim and Englert, 2004)). Another knee is predicted to occur at greater depth ( $\sim 10^4$  g cm<sup>-2</sup>,  $\sim 37$  m), where the production of TCN by slow muons ( $\Lambda_{\mu s} \sim 1500$  g cm<sup>-2</sup>) is overtaken by fast muons ( $\Lambda_{\mu f} \geq 4500$  g cm<sup>-2</sup>). This deeper  $\mu_s/\mu_f$  knee is more subtle than the  $n_f/\mu_s$  knee (attenuation ratio of only 3 vs 10). If surface erosion is negligible, then knees can be observed at these depths by measuring concentrations of TCNs. These concentration knees 'migrate' to shallower rock depths as the rocks exhume. Thus, the depth of the knees is indicative of erosion rate. However, at moderate or rapid erosion rates, the  $\mu_s/\mu_f$  knee may become unresolvable with current TCN measurement precisions (see below). Furthermore, our simplified calculations of the change in muonic flux based on synthetic TCN concentrations (Fig. 3) are insufficient in nature for many reasons, most important of which is the assumption of an ideal rock mass (flat-topped, homogeneous) and a simplified equation for a two-component muogenic production system. A more robust computation (discussed below) involving the probabilities of numerous interactions across varying

185

190

165 muon energy spectra with slant depth distributions based at least on the modern topography is necessary to compute the position of the knees and their response to irregular geometries of landscape evolution. To delimit the depth of the upper knee with measured TCN concentrations, we recommend a minimum of five samples over a 2 m depth to define the upper knee (the general position of this upper inflection can be approximated with current methods, Balco, 2017) (Fig. 3). If vertical core samples are used, it is necessary to have sufficiently high concentrations of quartz—or other mineral depending 170 on TCN—in the rock sample, otherwise a much longer core length will be needed per sample, eliminating the opportunity to tightly define the knee geometry (increasing horizontal sample width may alleviate this issue). Currently, the position of the lower knee is not sufficiently well resolvable to predict the location of the lower knee within a 10 m core length under a nonideal landscape, so more samples will be necessary. Furthermore, considering that the geometry and depth of the knees vary with erosion rates, unless erosion is known to be constant and on a flat surface, it will be necessary to oversample even for the upper knee.

Figure 3. Illustrations of variation in TCN concentration with depth. A: The concentration for <sup>10</sup>Be after 8 Myr with no erosion (left) and with a constant erosion rate of 20 m/Myr (right). B. Concentration vs log-depth profiles for <sup>10</sup>Be concentration, showing a minor, possibly resolvable position of the two concentration knees (arrows) for which theoretically will shallow with erosion. C. Computed production curves for different erosion rates at Sudbury, ON, based on Lifton (2014, LSD code) using geomagnetic field parameters over the past 8 Myr, and muon attenuation fitting constant  $\alpha = 1$  (Balco, 2017). Every fifth concentration point is plotted to show fitness of each curve fit (tolerance is within 2%, most points are buried under curve). An inflection point on each curve was determined with a two-componentcrossing method, assuming the data are strongly influenced by energetic nucleon and slow negative muon concentration profiles with exponentials containing attenuation lengths within a substantial pre-defined range.

Why consider deep muogenic nuclides (µTCN)? There are advantages at greater crustal depths to establish bedrock erosion rates over long time periods. First, TCN concentrations in shallow (metres) crust will be dominantly produced by fast nucleons with shorter attenuation length than muons, and therefore a narrower production cone of 45° (relative to muons) will collect







most (>80%) of the atmospheric and shallow crust spallation products (Gosse and Phillips, 2001, §3.4). The narrow production cone bounds a smaller horizontal surface area (smaller radius from zenith) and therefore is very sensitive to the history of burial within less than a hectare directly above the sample. At deep depths (hectametres), production is dominated by fast muons, and their >30x longer attenuation lengths and requirement for higher energies to penetrate to this depth means that the angular dependence is weaker, requiring a production cone >75° to capture >80% of the flux a deep sample receives (muonic flux angular dependence is site specific and very dependent larger scale topography). The combination of deeper depth and wider production cone means the bound horizontal surface area at ground level will be much larger (Fig. 4), diluting the impact of hectare-scale spatial perturbances at the surface (Fig. 5). The angular dependence of muons varies significantly with energy, so cones of 75°, 85°, and 89° are used to illustrate that at, for instance, a 200 m sample depth, the temporal or spatial variations in rock density, topography, and surface cover may be affected at horizontal ground-level radii of 10 km, over 1000 km<sup>2</sup> (Fig. 4). Hence small changes to  $\Delta S$  mass depth in some small portion of that  $2\pi$  hemisphere above a deep sample, i.e. ephemeral shielding by terrace gravels or water and ice) or deviations in bulk density (loess, snow, ash or colluvial cover, impacts of biological and soil processes, and temporal variability in these) have less an impact than the same variations above a shallow sample, unless the variations are spatially uniform over the entire horizontal surface area above the sample. If an esker, landslide, or narrow terrace gravel covered a sample location until a subsequent glaciation, the deep µTCN measurements would be weakly affected compared to measurements in shallow samples. Figure 5A compares the TCN concentrations expected in crust that was never covered, with concentrations under a permanent observable overburden thicknesses and densities. This would be the situation if the production cone surface above a vertical core sample was considered perfectly flat and horizontal, but any overburden was ignored. The case of no overburden above a sample is not plotted on Fig. 5A but would have 0% effect. In all overburden scenarios, the concentrations in the underlying samples (depth from top of overburden) will be greater because of lower mass depth shielding ( $\rho_{\text{overburden}} < \rho_{\text{crust}}$ ). Thinner or higher density materials (approaching  $\rho_{\text{crust}}$ ) will have less effect on the concentrations in the underlying crust. The curves for the shallow (1 to 10 m) covers illustrate that concentrations in deeper samples (µTCN) will be less impacted by surficial covers than samples where production is dominated by the energetic nucleon and slow muon fluxes (upper 30 m). The calculations also reveal the necessity to consider the thicknesses and densities of different overburdens, especially those with depths decametre thicknesses. Over cyclic timescales >10<sup>4</sup> yr or in rapidly denuding landscapes, such overburdens will not likely be permanent, so the magnitude of the effects indicated in Fig. 5A are maxima. To illustrate the effect of past overburden cover that no longer exists (i.e. crust is now at the surface), Fig. 5B shows the sensitivity of TCN concentrations with depth in crust if an overburden existed for the first 10% or 50% of the duration of exposure (erosion is zero, except for the instantaneous removal of the overburden; e.g. the lake drains, glacier melts, gravel is stripped). In all scenarios the TCN concentrations in the deeper samples are less sensitive to unknown previous overburden cover.

Figure 4. Geometric consideration of the area bound the production cones for energetic nucleons (upper few m) and muons (deeper depths). A wider cone angle or deeper sample depth will increase ground-surface radius (m, R 45°, 75°, 85°, 89°) from the zenith vertically above a sample (dashed curves, x10<sup>4</sup> m) and the horizontal surface area (solid curves, km², SA 45°, 75°, 85°, 89°) bound by the cone at the ground surface. This may provide guidance when selecting a core location and sample depths.




Figure 5. Effects of surficial covers on TCN concentration with depth. Relative, normalized concentrations are based on surface production at sea-level and high latitude with no erosion, for a short duration of 1000 yrs to avoid modelling effects of decay (outputs are approximately applicable to <sup>3</sup>He, <sup>10</sup>Be, <sup>21</sup>Ne, <sup>26</sup>Al), using attenuation lengths of 135, 1500, and 5000 g cm<sup>-2</sup> for fast neutrons, slow muons, and fast muons, and assuming slow and fast muogenic production rates are each 1% of total production at the surface. Densities (g cm<sup>-3</sup>) are crust: 2.70, gravel: 2.10, water: 1.00, ice: 0.92. A. Effect of existing overburden (%) on the TCN concentration measured in the underlying crust, if there is a permanent surficial cover with observable thickness and density over the entire horizontal surface area of the 2-π cone for each depth. TCN concentrations within surficial layers are not shown. Zero cover (100% crust) would have 0% effect. B. Sensitivity to past cover that is no longer observable. Assuming crust is at the modern surface, the curves provide representative scenarios that may occur in a glaciated region (e.g. southern central Canada). If there was no shielding, or the shielding occurred earlier than >6 mean lives of a radionuclide, the TCN measurements in crust would represent 100% of the expected production. In all scenarios, the overburden temporarily covered the entire cone horizontal surface at modern crustal depth 0 m. Erosion is zero except for the instantaneous removal of the overburden. One scenario (red dashed line) evaluates sensitivity to 50 m gravel cover that existed for the first 10% of the exposure time. All other scenarios assume a cover for the first 50% of exposure time.

A second advantage to using  $\mu$ TCN considers their relative insensitivity to variations is geomagnetic field intensity. At depths >50 m, the  $\mu$ TCN concentrations will be less affected by perturbations in the secondary cosmic ray flux owing to temporal and quasi-stationary spatial anomalies in the geomagnetic field, because to reach such depths muons require kinetic energies

which exceed geomagnetic field cutoff rigidities ( $Rc_{polar} \sim 0\text{-}1 \text{ GV}$ ,  $Rc_{mid\text{-}lat} \sim 3\text{-}10 \text{ GV}$ , and  $Rc_{equatorial} \sim 12\text{-}17 \text{ GV}$ ). Assuming rock and water densities of 2.65 and 1 g cm<sup>-3</sup>, atmospheric thickness of 1030 g cm<sup>-2</sup>, radiative loss factor of  $\sim 4 \times 10^{-6} \text{ cm}^2\text{g}^{-1}$ , ionization constant  $\sim 2 \text{ MeV cm}^2\text{ g}^{-1}$ , we estimate the energy loss of  $\mu_f$  at different depths to recover the primary kinetic energies of the muon in the atmosphere, and compare those energies to various cutoff rigidities of the geomagnetic field. Even at 10 m depth (0.0265 km water equivalent, km.w.e.) and at the equator, the median (50%) of muon energy spectra required at creation in the atmosphere significantly exceeds the Rc (Table 1). These geomagnetic advantages are in addition to the benefit of collecting multiple samples along a *horizontal transect* where any geomagnetically induced variation in the muogenic flux with time will be mostly irrelevant as it is systematic among all the horizontal samples.



Table 1. Estimates of cutoff energies at depths dominated by muon production

| Depth | Depth    | 50% of Eµ        | Eμ at creation |
|-------|----------|------------------|----------------|
| (m)   | (km.w.e) | at surface (GeV) | in atm (GeV)   |
| 10    | 0.0265   | 32               | 34.2           |
| 50    | 0.0325   | 35               | 37.3           |
| 100   | 0.2650   | 50               | 52.4           |
| 250   | 0.6625   | 80               | 82.4           |





There are other advantages to sampling at deep depths. Third, the concentration of a  $\mu$ TCN at >100 m depth is practically invariant over short distances (~10 m). This means that in a vertical transect over a relatively short distance, concentrations in three samples within a 10 m section of core will be equivalent within the  $\leq$  2% analytical precision, so they can serve as replicates, insensitive to small changes in density down core. Fourth, the TCN production rate at shallow depths is orders of magnitude greater than at hectametres. For this reason, shallower samples can provide higher resolution of erosion rates, i.e. during slow (mm-scale) erosion events over 10<sup>4</sup> yr or shorter timescales (e.g. glacial abrasion or grusification), only shallow samples will be sufficiently sensitive to precisely record the evolution. However, on cyclic scales of >10<sup>4</sup> yr or where thicker erosion losses occur (m-scale and larger, including episodic events such as glacial quarrying or rafting or cavitation in bedrock stream beds), the shallow TCN concentrations will be controlled by more recent exposure history and require additional knowledge for treating cases of episodic erosion (e.g. Margreth et al, 2015). However, the relatively muted signal in the deeper  $\mu$ TCN concentrations can treat those same m-scale events as gradual steady erosion (sensu Lal, 1991), not episodic—an assumption better suited to TCN erosion rate calculations. Thus while the  $\mu$ -paleotopometry method has limitations, it seems advantageous for questions involving long-term landscape evolution.






In summary, subsurface samples which have less shielding today than a million years ago owing to surface erosion, will have a predictably lower  $\mu$ TCN concentration than expected for their modern depth. For an estimate of the time-constant absolute rate of erosion at a location on Earth using samples from a near vertical core (*vertical \mu-paleotopometry*), we can compare the measured concentration in the rock with the theoretical concentration based on nearby deep muon monitoring or the GEANT-4 computation. It may also be possible to exploit the depth of  $n_f/\mu_s$  and  $\mu_s/\mu_f$  concentration knees for independent constraints on erosion rate. For an estimate of relative erosion (i.e. relief production), we can compare the  $\mu$ TCN concentrations measured in samples along an underground horizontal transect (*horizontal \mu-paleotopometry*) below peaks or valleys. Over time the samples under a deepening valley will have proportionally higher concentrations than samples under interfluves. The concentration disparities will depend on initial relief, duration and rate of relief production, and spatial variability of that incision. For either the vertical or horizontal approach, the concentrations can be used to explicitly compute the erosion rate if it is assumed constant with time. The variation among concentrations of multiple  $\mu$ TCNs may be useful to approximate the amount of shielding loss ( $\Delta$ S) due to *variable* erosion rates.

#### 295 3. μ-paleotopometry strategies

Applications, advantages, limitations, and examples of sampling and computational strategies for *Horizontal* and *Vertical*  $\mu$ -paleotopometry are explored next. While we focus on erosion (denudation or incision), the method may also be used to provide rates of thickening above a sample.

# 3.1 Horizontal µ-paleotopometry

Horizontal  $\mu$ -paleotopometry is appropriate where the timing and *relative* rate of incision across a landscape is sought, specifically to quantify the rate and timing of relief generation. This may be negative relief such as stream incision or fjord development, or positive relief such as creation of an anticline or transpressional ridge. By measuring concentrations in samples collected along a deep subsurface horizontal transect, there are possibly two less degrees of freedom in the shielding equation. First, all samples are collected at nearly same time-averaged depth, so their initial pre-incision concentration will be similar if the terrain initially was relatively flat. Second, temporal variations in the  $\mu$  flux (and production rate of  $\mu$ TCN's) owing to geomagnetic or GCR variations will be similar for all samples and may be ignored for horizontal  $\mu$ -paleotopometry. The latter advantage is important because it reduces the impact of our current uncertainty in muogenic production rates at depth. In a related way, horizontal  $\mu$ -paleotopometry is particularly applicable where ice sheet cover is expected but temporal changes in thickness are unknown, because the impact of the ice cover history will be uniform across all samples. Thus the differences in concentration across the relief (valley or hill) are essentially a function of  $\Delta$ S in the crust.

Sampling may be accommodated by existing road or train tunnels, mine tunnels, or a series of vertical drill cores to similar depths. The number of samples required depends on the nature of the question and ideally samples will extend under the valley





of interest and adjoining interfluves. The simplified illustrations in Figure 6 represent two 'end-member' scenarios of valley incision. In the scenario on the left (Fig. 6A), the incision of a flat plane began sometime after 8 Myr BP and continues. Over this time the pattern of <sup>10</sup>Be concentration in quartz along the deep horizontal transect would predictably reflect the shape of the  $\Delta S$  above the sample transect. In this scenario, the measured concentrations under the valley will be less than that predicted if the valley was formed 8 Ma ago. The concentration of each sample in the transect will be affected by the three-dimensional topography above it (i.e. the shape of the surface bound by the production cone did not remain horizontal), so will not be simply an exact mimic of  $P_{AS}$  vertically above each sample as illustrated (discussed below). Average time-constant erosion rate can be computed for each sample and may provide insights into how the incision progressed (spatial and temporal changes). Any history before 8 Myr would not be recorded by <sup>10</sup>Be (half-life 1.39 Myr, (Chmeleff et al., 2010; Korschinek et al., 2010)); a longer-lived or stable µTCN would be needed for that (e.g. 21Ne in quartz with minimum pathways for nucleogenic and radiogenic <sup>20,21,22</sup>Ne). By choosing the optimal combination of isotope(s) and sample depth (further discussed below) it may also be possible to establish constraints on the long-term average fill depth of a basin or ice shielding in a fjord (positive relief). The example on the right depicts the unlikely situation that a valley with the same geometry was rapidly carved into a negligibly eroding plateau only 5 kyr ago (Fig. 6B). Over such a short term, the concentration of <sup>10</sup>Be under the valley may not record any  $\Delta S$  as its decay rate and production rate are too slow. A shorter-lived isotope (e.g.  $^{26}$ Al or  $^{36}$ Cl) may record  $\Delta S$  for the more recent past, and for a 5 kyr event it may be necessary to use muogenic <sup>14</sup>C. Thus, using only <sup>10</sup>Be or <sup>21</sup>Ne and assuming a constant erosion rate, it may be possible to estimate when incision began and the average rate of relief generation over timescales >10<sup>4</sup> yr. If two or more isotopes with different decay rates are measured in each sample in the horizontal transect, a time-varying  $\Delta S$  and paleotopometry may be achieved.

Figure 6. The pattern of deep μTCN concentrations along a horizontal transect reflects the history of changing  $\Delta S$  relative to a reference frame of horizontal μ-paleotopometry. The cross-sections depict end-member scenarios for a long-lived or stable isotope (e.g.  $^{10}Be$ ). A. The valley incision began within the saturation period for  $^{10}Be$  at very low erosion rates (ca. 8 Myr of exposure) and the plateau was horizontal and flat prior to this. Yellow polygons represent ephemerally stored sediments which appear and disappear during the incision. B. The valley was created in the past 5 kyr, which is too recent for the low  $^{10}Be$  concentrations at depth to capture it. Most incised valleys will have a horizontal transect concentration profile between these two end-members.







#### 3.2. Vertical µ-paleotopometry

In instances where an absolute constant long-term average change in crustal thickness (erosion or thickening) at a specific location is sought, or where there is no access to a subsurface horizontal transect, samples can be collected in a vertical transect (Fig. 7) along for instance a vertical core, a sub-vertical core, or a tunnel wall. As mentioned above, if lithological densities and topography differ significantly among the production cones for each sample (and their cone bound surface areas, this must be considered (Fig. 4). The Vertical transect strategy has two approaches, one relying solely on the μTCN concentrations compared to theoretical concentrations for modern  $\Delta S$ , and the second involving a comparison of the muon flux based on muon monitor data collected at deep underground neutrino labs and the paleo-muon flux required to explain the measured µTCN concentrations. Below 50 m, where the TCN production is dominated by fast muons (possible exceptions include thermalneutron capture), the advantage to sampling multiple samples downcore is simply to improve statistics. Deeper samples will be less sensitive to ephemeral changes to shielding (e.g. by thin gravel cover or water, Fig. 5). However deeper samples may have the disadvantages of poor AMS precision and of a large production cone that spans across potentially undesirable topographic or compositional changes. Two additional tactics should be considered when using the Vertical transect strategy. The first is to determine the depth of one or both concentration knees in the TCN concentration with depth, as discussed above (Fig. 3): this may provide a more recent erosion history than the deeper fast muogenic TCN. The second tactic is to compare the measured concentration to a concentration that is sufficiently shallow to allow the measurement precision (discussed below) needed to compute erosion rates at the required resolution, but sufficiently deep (Fig. 7) to minimize impacts of ephemeral changes in shielding thickness (e.g. ice, water, sediment, Fig. 5). While it is possible to approximate  $\Delta S$  and surface erosion rate with just one subsurface sample, multiple closely spaced samples will improve resolution. Furthermore, multiple µTCN measured in each may be useful to detect time-varying erosion rates for evaluating tectonic, isostatic, or climate-induced landscape changes (Fig. 7). For instance, in an area subject to Quaternary ice sheet erosion, <sup>10</sup>Be would be expected to exhibit a much slower mean constant erosion rate (over Pliocene and Quaternary) than <sup>26</sup>Al in the same quartz samples, because <sup>26</sup>Al decays faster so only captures erosion mostly over the Quaternary period.




Figure 7. Illustration of the sampling strategy for μ-paleotopometry using two radioisotopes with different decay rates in samples below 50 m rock depth. The coloured curves are isoeroderes for constant erosion rates ranging from 0 to 50 mm/kyr (m/Myr), calculated using only the fast muon Matlab® code provided with (Balco et al., 2008) online calculator, but using updated LSDn constants and geomagnetic field inputs for 2 Myr of production and beyond (Lifton et al., 2014) and muon attenuation fitting constant α = 1 (Balco, 2017). The wider spacing toward the top provides the improved resolution of erosion rates.

#### 5. Proofs of concept and sensitivity analyses

Two first-order challenges remain. Can we measure the concentrations of  $\mu$ TCN at extreme mass-depths with precision suffice for establishing the erosion rate? Can we meaningfully resolve a time-varying erosion rate history, or are we restricted to a time-constant erosion rate estimate? Here, we provide the state of knowledge on these questions.

#### 5.1. Measurement precision

A first order limitation is the difficulty in measurement of low concentrations of stable (<sup>3</sup>He, <sup>21</sup>Ne) or long-lived radionuclides (e.g. <sup>10</sup>Be, <sup>26</sup>Al, <sup>36</sup>Cl, <sup>41</sup>Ca). AMS measurement precision on a BeO target with a known abundance of <sup>10</sup>Be atoms can be predicted by estimating the total counting efficiency (Fig. 8). For example, assume a 100 m deep sample has a concentration of 480 <sup>10</sup>Be atoms/g quartz. The *optimal* precision of a 1 g sample of that quartz (dashed curve, assuming 1.1% total counting efficiency, i.e. if the cathode has high BeO purity, packed properly, and run to exhaustion) is ± 61.5% (1σ) on the current configuration of the CAMS-LLNL AMS. *Typical* counting efficiencies are less (solid curve, 0.55%) yielding an estimated precision of ±113% for the same gram of quartz. To acquire a 10% precision on that quartz, we would need 38 g or 156 g quartz for optimal or typical counting efficiency, respectively. Lower depths (lower concentrations) will require greater quartz mass. Unfortunately, greater quartz mass requires more acids for digestion and longer evaporation times, potentially elevating process blanks which will approach the same order of magnitude of measured atoms. To obtain a sufficient TCN abundance for the desired precision, planning must begin before sampling to establish the minimum quartz mass (and therefore rock mass, length of core) necessary to obtain a sufficient TCN abundance. As the purity and loading of the target also become very





important at these TCN abundances, communication between the target prep lab and AMS facility during the planning stages is necessary. In the SUFCO Mine, Utah, horizontal  $\mu$ -paleotopometry experiment in 2018 described below, we obtained 17-27% precision (see below) on  $^{10}$ Be from quartz samples at 183 to 250 m depth ( $^{10}$ Be concentrations ranged from 296 to 494  $^{10}$ Be atoms/g qtz). The outlier sample with high (27%)  $1\sigma$  AMS precision was likely caused by insufficiently cleaned quartz, poor target chemistry, or poor target loading including non-homogeneous mixing with Nb. In a recent experiment on a subvertical core extracted near Sudbury, Canada, measurements of cosmogenic  $^{36}$ Cl at 250 m depth were obtained with AMS precisions of 3% - 4% ( $1\sigma$ ) on potassium feldspars, in part because a high total-Cl concentration uniformly through the core provided thermal-neutron capture produced  $^{36}$ Cl, improving counting statistics (McLellan, in prep). Precisions attainable on total Cl ( $\sim$ 1%) and relevant radioactive parents (3%) indicate this  $\mu$ -paleotopometry TCN will also be valuable for evaluating erosion rates over the past Myr.

Figure 8. The expected precision of <sup>10</sup>Be/<sup>9</sup>Be measurements on the FN at CAMS-LLNL, modified from (Hidy et al., 2018). Solid line assumes optimal total counting efficiency (1.1%) and dashed line is typical counting efficiency (0.55%). The typical efficiency may be used to estimate the sample mass of quartz needed to provide the desired precision. The orange dots indicate the total precisions reported to 1σ confidence, for the measurements of <sup>10</sup>Be in each SUFCO sample for which the total analytical precision at 1σ confidence includes AMS error in the measurement of the radionuclide to stable nuclide ratio (R/S), uncertainty in carrier concentrations, and the impact of the uncertainty in the measurement of R/S in the process blank). At low atomic abundances, the AMS precision contributes the greatest uncertainty.

# 5.2. Horizontal $\mu$ -paleotopometry on the Colorado Plateau.

To determine if measurements of low concentrations of  $\mu TCN$  at  $200 \pm 100$  m depths have sufficient precision to ascertain a long-term erosion history, we conducted a proof-of-concept study, where a single valley likely formed within the past 8 Myr,

and where the interfluvial plateau was relatively flat and horizontal ( $\leq 6^{\circ}$ ) and showed no expression of complex erosion. The 420 location has no significant probability of glacial ice cover in the Quaternary. The site selected is a tributary of the Colorado River, in the Book Cliffs region 50 km NE of Salina, Utah (Fig. 9), where the SUFCO Coal Mining company agreed to provide samples along a horizontal underground mine adit through sub-horizontal quartz arenites of the coal-bearing Upper Cretaceous Blackhawk Formation. Six quartz arenites samples were collected from above the Upper Hiawatha Coal Seam (Fig. 9, 425 Supplementary Data Table S1). The sample material consists of sandstone sourced from crevasse splay deposits with an average porosity of 20 - 30%, and a bulk density of  $2.20 \text{ g cm}^{-3}$ . The mine stope runs laterally under the incised valley along a ~N-S trend at an elevation of 2343 m, a depth of ~150 m below the lowest portion of the incised valley (Fig. 10). Sample 1 was directly under the western interfluvial plateau (Fig. 9, 10) and the other five samples were under the valley, with samples 4 and 5 straddling the deepest part of the valley. The six SUFCO samples consist of arenite sandstone material with the minor 430 presence of Fe- garnet, hornblende, and potassium feldspar, as well as coal micro fragments, which comprised a substantial volume of the sample even at fine grain sizes. Generally, the quartz percentages ranged from 80% to 45% in the original samples. The samples underwent partial physical processing (crushing, grinding, sieving, pre-rinsed and partially leached in aqua regia) at Lawrence Livermore National Laboratory (LLNL) and at CRISDal Lab, Dalhousie University, mineral separation (see Supplementary Data §S1, Table S2). For all samples, sand aliquots of each sample we placed in cleaned alumina 435 vials and combusted at 450°C (minimum of 5 hr in muffle furnace) to further break the quartz grains to access and remove the carbon. Target preparation for BeO was completed at CRISDal Lab (§S2) and the measurement of <sup>10</sup>Be/<sup>9</sup>Be by accelerator mass spectrometry was conducted at CAMS-LLNL. The reduced data (Tables S3 and S4) are the blank-subtracted concentration of <sup>10</sup>Be per gram of quartz in each sample (Table S4) and the uncertainty in that concentration.

**Figure 9**. SUFCO Mine proof-of-concept experiment. DEM and block diagram showing the geometry of the incised valley and the relative location of the samples collected in a horizontal adit in the Blackhawk Formation, SUFCO Mine, Utah (modified from (Soukup, 2018)).

Figure 10. SUFCO Mine proof-of-concept experiment. Lower panel: Topographic profile of the proof-of-concept study site for the horizontal  $\mu$ -paleotopometry method, showing the relative position of the sub-surface samples along a 600 m horizontal transect. The greatest relief of the valley relative to the interfluvial plateau on the left is 112 m. Relative position of the samples (blue dots) where collected in the adit (18 m height). The true depths of each sample below the modern surface were provided by SUFCO Mine. Middle panel: Measured  $^{10}$ Be concentrations (red dots), with total analytical precision (1 $\sigma$  confidence). Top panel: Direct interpretation of concentration as erosion rate







(black dots), assuming vertical flux (i.e. no adjustment for changing in muon flux as absorption mean free paths evolve with topography in all directions above each sample).

While the experimental setting seems relatively simple, there are assumptions that must be verified, avoided by using a purposeful sampling strategy, or considered in terms of the final uncertainty in the erosion rate estimate. For instance, the western plateau appears to be relatively stable, owing to the presence of resistant near-horizontal local caprock in the Blackhawk Group. However, the plateau surface on the east has some relief indicative of differential erosion by the drainage or complex erosion along tributary gullies. We cannot independently confirm when or where the incision of the main channel started, or that the incision remained symmetrical in cross section over time, or that incision rate was constant in space or time.

The <sup>10</sup>Be concentrations in all six samples were calculated from the AMS results (Table S4). The sample masses obtained were <250 g. During the target preparation in 2018, sample SUFCO-006 was not sufficiently pure to create a target (several major elements were > 100 μg/g), so only five samples were prepared and analysed in January, 2018. A duplicate of sample SUFCO-001 (entirely from fresh quartz, not exact split) was also measured to test reproducibility. The 10Be concentration calculated for the quartz samples ranged from 296 to 1119 atom/g (Table 2). The duplicate samples had concentrations within their 1σ measurement error. The ratios of <sup>10</sup>Be / <sup>9</sup>Be for the samples averaged only 1x10<sup>-15</sup>, except samples SUFCO-004 and -005 which were 2 and 4 times higher (Table S4), and the process blank subtraction ranged from 4% to 14% of the measured atoms (Table 1). During this AMS analysis the pure standards used for normalization consistently obtained mean of the stable isotope (9Be) of 24 µA on the CAMS LLNL 10-MV FN accelerator. The natural samples had lower currents, presumably owing to impurities in the target material (Table 1) and contributing to weaker counting statistics. While currents of 5 µAmps and higher can yield meaningful results with reasonable counting statistics, SUFCO-004 and -005 had very low currents (2.9 and 1.9 µAmp) and vielded lower precision results (31% and 30% respectively, Table S4) despite having higher concentrations. Therefore in 2019, samples SUFCO-004 and 005 were purified further, and their targets analysed with a SUFCO-006 which by that time also had purer quartz (as indicated by Al concentrations well below 100 ppm). After those three targets were measured on Nov 22, 2019, the concentrations of SUFCO-004 and -005 were calculated to be significantly lower (Table 1) with an AMS uncertainty similar to all other samples (~17%), and all three samples had currents that were similar to targets SUFCO-001, 002, 003, and 001dup in the 2018 run. On this basis, we ignore the 2018 measurements of SUFCO-004 and -005 when interpreting erosion history, and we have taken a mean of the two SUFCO-001 duplicate targets and assumed the greater relative  $1\sigma$  error of 17%. Note that the lines of equal erosion rate (isoeroderes) converge with depth, but the error bars are relatively consistent, indicating an advantage of measuring the µTCN at shallower depths.

Table 2. Summary of measurements for SUFCO-Mine proof-of-concept experiment

| Field ID            | Atoms in process blank | Atoms in sample <sup>1</sup> ( <sup>10</sup> Be atoms) | Blank<br>fraction <sup>2</sup><br>(%) | Concentration                   | 1σ Unc | σ Unc <sup>9</sup> Be current <sup>3</sup> |       |
|---------------------|------------------------|--------------------------------------------------------|---------------------------------------|---------------------------------|--------|--------------------------------------------|-------|
|                     | (10 Be atoms)          |                                                        |                                       | ( <sup>10</sup> Be atoms/g qtz) |        | μAmp                                       | CoV%⁴ |
| First Measurements  |                        |                                                        |                                       |                                 |        |                                            |       |
| SUFCO-001           | 4.367E+03              | 2.117E+04                                              | 11%                                   | 423.6                           | 78.38  | 11                                         | 10%   |
| SUFCO-002           | 4.367E+03              | 1.850E+04                                              | 12%                                   | 369.2                           | 67.07  | 15                                         | 5.8%  |
| SUFCO-003           | 4.367E+03              | 1.488E+04                                              | 14%                                   | 295.5                           | 79.95  | 7.2                                        | 7.0%  |
| SUFCO-004           | 4.367E+03              | 3.514E+04                                              | 7%                                    | 703.1                           | 223.7  | 2.9                                        | 6.8%  |
| SUFCO-005           | 4.367E+03              | 5.583E+04                                              | 4%                                    | 1119                            | 337.2  | 1.9                                        | 5.5%  |
| SUFCO-001 DUP       | 4.367E+03              | 1.697E+04                                              | 13%                                   | 339.4                           | 71.07  | 10                                         | 5.3%  |
| Second Measurements |                        |                                                        |                                       |                                 |        |                                            |       |
| SUFCO-004 REP       | 1.15E+04               | 2.263E+04                                              | 5%                                    | 451.9                           | 76.21  | 7.6                                        | 1.1%  |
| SUFCO-005 REP       | 1.15E+04               | 2.475E+04                                              | 5%                                    | 494.0                           | 90.25  | 5.9                                        | 7.0%  |
| SUFCO-006           | 1.15E+04               | 1.689E+04                                              | 7%                                    | 337.3                           | 70.11  | 5.9                                        | 8.4%  |

- 1. Atoms in sample (corrected for blank subtraction)
- 2. Process blank atoms divided by uncorrected measured atoms in each sample
- The °Be current averaged over the first three runs (some targets lasted longer, but all lasted for at least three)
   Coefficient of variation about the mean current, calculated as standard deviation in currents over the three runs divided by mean current




The concentrations of  $^{10}$ Be $_{\mu}$  of six samples from depths between 176 and 250 m can be interpreted as time-constant erosion rates (Fig. 10, 11). In this overly simplified approach, we are attempting only to demonstrate feasibility. To estimate the relief generation, we are assuming that the subsurface sample had a flat surface above and around it (i.e. we have not adjusted for the 3D changes in the relief). The only sample under the western interfluvial plateau (SUFCO-001) where there is no significant relief generated, indicates the slowest landscape erosion rate of 10 mm/kyr. This is reasonable for a slow semi-arid region with shallow gradient above moderately competent sedimentary rock. The erosion rates of the other samples are more than double this. SUFCO-002, which has the next lowest relative shielding change, yielded the second slowest erosion rate. The fastest erosion rate is indicated by SUFCO-003. This may be a result of uncertainty in the measurement, may reflect a spatio-temporal variation in erosion history where the shielding above SUFCO-003 eroded faster than the others in the past million years, or may be the result of a combination of these factors plus geometrical variations in the 3D surface relief (today's relief may not represent the time-averaged geometry). We note that the 50 mm/kyr isoerodere curve is within the 1 $\sigma$  uncertainty of all samples except SUFCO-001. The difference between the erosion rate of the interfluvial plateau and the valley (i.e. 10 and ~50 mm/kyr) provides an estimate of valley relief production. At 40 mm/kyr, it would have taken 2.8 million years assuming constant incision. Considering that this valley tributary may be a result of headward erosion since the uplift of the Colorado Plateau, which has caused incision of the Colorado River in western Arizona beginning around 7 Myr ago, and that there are many studies that have argued for increased erosion in the Quaternary (last 2.58 Myr), this date for initial incision based on the incision rates calculated seems plausible. A more



510 rigorous evaluation of the change in muon flux to a subsurface sample during an evolving 3D landscape will improve this interpretation.

515 **Figure 11.** Depth vs. μTCN concentration graph. Samples with greater erosion rates plot to the left. Solid black dots are <sup>10</sup>Be concentration in quartz for six SUFCO mine samples. Uncertainties are 1σ total analytical error (AMS counting statistics, carrier concentration uncertainty, uncertainty contributed from the process blank subtraction). For sample location with respect to the incised valley refer to Fig. 10. The isoeroderes assume time-constant erosion, for 0, 1, 2, 10, 20, and 50 mm/kyr erosion rates, and represent the saturation concentrations with  $t = \infty$ .

# 5.3. Horizontal μ-paleotopometry under Swiss Alps.

A second proof of concept study underway (Raab et al., 2024) uses deep subsurface samples retrieved from the Ceneri Base Tunnel (CBT), a dual rail track tunnel in Cantone Ticino, in southern Switzerland, permitting access to crystalline basement and metasediments in the central Alps. The Swiss government generously provided co-author GR the core samples which contained sufficient coarse quartz to conduct a horizontal  $\mu$ -paleotopometry experiment. The samples were housed for ~12 years in a thin-roofed storage facility Switzerland until 2020, when the samples were shipped to Dalhousie University and they were stored under 8 stories of the Life Sciences Centre since. In October 2022 the  $^{10}$ Be concentration in quartz from the first of the samples (GBC-SE-01, with a modern depth of ~140 m, Fig. 12) was determined to be  $370 \pm 30 \ (1\sigma)$  atoms  $^{10}$ Be/g, using 50 g of quartz. Its total analytical uncertainty is 9.6%  $1\sigma$ , about half of the uncertainty in the SUFCO Mine samples measured five years earlier (Table 2). Relative to the SUFCO experiment, a chemically purer quartz separate was achieved, and the CBT sample was shallower. At face value, the corresponding erosion rate assuming a vertical flux (i.e. no correction for topography, density, 3D effects on muon flux) is >100 m/Myr.

**Figure 12.** Longitudinal geological profile for the Cereni Base Tunnel (Merlini et al., 2018), southern Swiss Alps, illustrating the coarse structural and lithological elements, and the approximate location of the first sample analysed (yellow dot) under Val Colla near the city of Vezia, Cantone Ticino.

#### 5.4. Consideration of the 3D shielding geometry and the muon flux to a subsurface sample.

Owing to the different angular dependence of muons to that of nucleon flux a larger portion of muons can arrive at very shallow angles to a sample through rock distances beyond 50 km (§2, Fig. 4). Computing the fully 3D-flux under a region with complex relief, with adjustments for energy-dependent slant depths (fluxes at different angles of incidence) while considering 3D variations in mass depth along each muon trajectory is the goal of the latest version of MUTE (3.0), so GEANT-4 can be provided the necessary probable particle flux energy spectra and muon slant depths. At the time of this paper, MUTE (3.0) is not yet able to incorporate complex density data beyond a 50 km radius from each sample which would be useful for samples below 250 m depth. However, to demonstrate the topographical impact on muon flux to a sample a hectometre below the surface, we computed the shielding factor for a sub-surface sample near the Sudbury Neutrino Observatory. Samples for the Sudbury vertical  $\mu$ -paleotopometry experiment from Core BH1470760 (deepest sample 1000 m), collected and donated in 2024 by Vale Inc's Morgan West exploration drilling project (Maclellan and Gosse, 2024) are being processed for  $^{10}$ Be,  $^{26}$ Al, and  $^{36}$ Cl. Owing to the relatively homogenous ( $\rho_{mean} = 2.69 \pm 0.01$  g cm $^{3}$ , based on replicate measurements on 17 core fragments from the surface to 250 m) granophyre mapped to be entirely within the 75° zenith cone to 250 m (compared to different clastic, carbonate, and coal units or metasediments and crystalline basement in the Utah or Swiss sites), we can isolate the effect of modern topography on flux to each sample (Fig. 13). This simplified computation currently ignores bathymetry below

lake surfaces (i.e. the topography is based on the DEM surface, which includes the hydrosphere). Even with these simplifications, an azimuthal variation in muon flux with depth and azimuth is apparent (Fig. 13) despite the moderate relief, characteristic of the southern Canadian Shield. Using MUTE (2.0), the total flux to a sample at 250 m is 5.9380 x 10<sup>-6</sup> muons per cm<sup>-2</sup> s<sup>-1</sup> (spanning a wide range of energies and only considering modern geomagnetic field model; uncertainty has not yet been established). The 3D-flux and accompanying energy spectra will be needed for computing deep TCN production with GEANT-4, and for computing aspects of paleotopometry. A sensitivity analyses of the impacts of variation in relief geometry over the timescale of interest will be completed once MUTE (3.0) is fully operational.

Figure 13. Progress on calculating the total muon flux and energy spectra to a sample, based on slant distance from surface topography to an underground sample along all possible muon arrival trajectories for an irregular relief. Top Left: Schematic of muon trajectories (red arrows) directed toward the underground sample (blue box) at a vertical depth of 250 m, at θ = 30° from zenith, and irregular surface topography. To compute slant depth, the MUTE v 2.0 code traces each trajectory from the sample upward until it intersects the surface defined by a digital elevation model (DEM). Trajectories are sampled across all azimuths (φ = 0–360°) at a fixed zenith angle and the process is repeated for a full range of zenith angles (θ = 0–75° is currently computationally achievable, 0° is vertical). Top Right: DEM of the Sudbury study area (~5 x 5 km grid), with the core location marked by a black star. Lower: 3D-intensity map of the muon flux. In studies involving a river valley above the samples, the asymmetry will be more apparent. Density was measured to be very uniform to 250 m (2.69 g cm<sup>-3</sup>).

# 575 **5.5. Time-varying erosion rate**

The previous proofs of concept assume erosion rate was constant over the entire exposure duration recorded by an isotope (e.g. 8 Myr for  $^{10}$ Be). It is unlikely that landscape erosion rates are constant at any timescale. To provide a preliminary assessment of the sensitivity of  $\mu$ -paleotopometry approaches to transient erosion rates, we invert synthetic TCN concentrations which are forward modelled from input transient erosion rates. These synthetic concentrations are inverted for erosion rate with a Markov

580 chain Monte Carlo algorithm to recover a distribution of erosion rates compatible with those concentrations. We then compare the distribution of inverted erosion rates with the input erosion rates.

Drawing inspiration from the Colorado Plateau example in §5.2, we use a simple synthetic relief geometry, assuming a final valley relief of 250 m incised into a plateau at 2500 m asl, with sample depths of 300 m below the plateau. The integral of the erosion rate over time must equal the relief with respect to the plateau elevation where, in this case, erosion is assumed 0 m/Myr—this constrains the total eroded thickness at the synthetic sample location. The input valley incision rates were 2 m/Myr from 8 to 4 Ma (BP), 8 m/Myr from 4 to 2 Ma, 20 m/Myr from 2 to 1 Ma, and 50 m/Myr over the last million years. The inverted valley incision rates range anywhere from 0 to 50 m/Myr (i.e. a priori erosion rate distribution) during the same four decreasing time intervals (Fig. 14). We use a simple exponential curve with vertical attenuation length 5700 g cm<sup>-2</sup> (i.e. in lieu of the Balco (2007) implementation of the Heisinger et al. experiments in 2000). This shortened computation time by removing the need for a forward Euler implementation and allowing implementation of Lal (1991, eqns 8 and 11, i.e. assuming steady state within each time interval). The inverse model uses the same forward model and temporal grid as that used to create the synthetic sample. Erosion rates are inferred using a Markov chain Monte Carlo (MCMC) sampling of the a posteriori probability distribution of the erosion rates in each time interval. The code is flexible—site specific parameters, time intervals, and erosion rates are specified via an input JSON file, all required python packages are free and open source, and can be deployed on a single pc or distributed environment. Once MUTE v3.0 with 3D-density correction is operational with GEANT-4, those production rate data will substitute for the simplified approach taken here.



**Figure 14.** Output from an MCMC inversion model to evaluate the ability of  $\mu$ -paleotopometry to resolve an average erosion rate from measurements of  $^{10}$ Be and  $^{26}$ Al at 250 m depth, if the natural erosion rate was allowed to vary during four different time steps over the past 8 Myr years.

Using a single isotope, a unique solution was not achieved—only long-term average (i.e. assuming constant) erosion rate can be resolved with a single isotope. If two or more isotopes are used, a solution is possible for a well posed problem (e.g. two nuclides and two unknowns). The inversion recovered the erosion rate for the most recent time interval (past 1 Myr): the maximum a posteriori probability erosion rate closely matches the actual (input) erosion rate (red line in the bottom histogram) and the inverted range is tight. For earlier time frames the a posteriori range is consistent (shown by the grey histogram in the left three panels) but the range is wide and the maximum a posteriori probability erosion rate is farther from the actual erosion rate than for the last Myr. This is likely due to an ill-posed problem – searching for four erosion rates with two isotopes. To








resolve transient erosion, a reduction in degrees of freedom is needed. This experiment does however demonstrate the sensitivity of this method to time-varying erosion and may be applicable to inferring landscape changes where sufficient constraints are available to give a well-posed problem (e.g. a non-glaciated area with reasonably flat initial surface, see §5.2). With measurements of  $^{10}$ Be and another  $\mu$ TCN and a reasonably steady, low background erosion rate on a plateau with incised valley, it is possible to recover the slow plateau erosion rate and the most probable timing of onset of the faster erosion within the valley (eroded thickness determined by relief, and the incision determined by relief and onsite timing). Complications such as large-scale variations in  $\Delta$ S above the samples can be considered (e.g. in an active mountain range where significant changes to topography occurs over 1-2 Myr timescales (e.g. Central Swiss Alps, §5.3), or where ice sheet or water cover thickness has varied greatly (§2, §5.4)), but the uncertainty in those factors would need to be propagated through the model efforts.

#### 6. Future improvements

The purpose of these proofs of concept and sensitivity analyses is to establish the feasibility of using µTCN to evaluate longterm erosion events. While these relative and absolute erosion rates and incision rates seem plausible, several improvements are recommended to enable the full application of μ-paleotopometry. Considering the low concentrations (~300 atom/g) of <sup>10</sup>Be in quartz at these depths, multiple targets (from aliquots of a sample) should be measured at similar depth to improve uncertainty in the concentration. A greater understanding of the spatiotemporal variability in erosion (the 3D variations) and attempts to evaluate the spatial evolution of a valley will require more samples along a horizontal transect than the SUFCO experiment. The μ-flux and energy spectra are sensitive to topographic relief. As discussed, a program such as MUTE (Fedynitch et al., 2022; Woodley et al., 2024) should be integrated into the flux and production rate calculation. However, to do this, it is necessary to compute the surface fluxes as a function of surface energy and surface incident angle (e.g. MCEq, (Fedynitch et al., 2015)), use those fluxes to compute the survival probability of muons to the subsurface sample (e.g. PROPOSAL, (Koehne et al., 2013)), and then compute the product rate of the µTCN of interest using that flux energy spectrum for each sample depth (e.g. GEANT4, (Collaboration and Agostinelli, 2003)). If density is not homogeneous, each incident muon path (azimuth, plunge) in PROPOSAL will have a mass depth that is unique even for its incidence angle. The result of combining these models will be an improved interpretation of measured concentrations of <sup>10</sup>Be as erosion rates, with 3D effects related to modern relief fully considered. Once these models are fully operational, additional sensitivity analyses will be needed to determine the tolerances of knowledge gaps with respect to spatial variation in rock density and time variation in relief geometry. This process can allow testing of different simple scenarios of the history of topographic change or a landscape evolution model. Currently, for very low or zero erosion rate (e.g. Beacon Hill site, Antarctica, Balco, 2017) it is possible to resolve the upper  $(n_f/\mu_s)$  knee. However, as the knee position shallows and deforms with erosion, resolving the knee with sufficiently high resolution may not be possible unless strategically targeted samples with sufficient abundances of the targeted minerals are analysed. Additionally, a significant improvement may be achieved if multiple μTCN are measured in the same samples. Differences in the decay rates of those μTCN may allow resolution of temporal







variations in the landscape erosion, a limitation of other paleotopometry methods. Improvements in target lab procedures and infrastructure, obtaining sufficiently large samples to permit even greater purification of the selected minerals, and reducing and stabilizing the background and process blanks at the time  $\mu$ -paleotopometry experiments continue to be undertaken and will be required if improved resolution or spatiotemporal variability in topography evolution is sought. For the ongoing work at the Sudbury site, the granophyric quartz has minute intergrowths which is preventing mineral purification so than any non-Si majors are 

precisions, and is responsible for all AMS measurements. LM contributed data and figures for Sudbury research. MS provided an early version of Fig. 9 and the supplementary material. GR provided an early version of the CBT summary and Fig. 12. MD provided the computational results in §5.5. SN and JP focused on sections pertaining to landscape evolution and relief generation. M-CP and WW contributed insights drawn from the dark matter community's advances in measuring and modelling muon fluxes with MUTE and PROPOSAL, and parts of §2 including Fig. 2.





# **Competing interests**

The authors declare that they have no conflict of interest. JCG has received funding from Canada's Nuclear Waste Management Organisation to evaluate the utility of  $\mu$ -paleotopometry for estimating long-term erosion rates in continental shield environments.

#### Acknowledgements

We thank Peter Hayoz from the Bundesamt für Landestopographie (swisstopo) for the possibility of taking samples from the Gotthard Base Tunnel (GBT) and Ceneri Base Tunnel (CBT) core archive. We are grateful for underground samples from the Utah site provided to AJH and JP by SUFCO Mine in 2016. MD and SN and the analysis of the CBT and Sudbury-VALE samples were supported by the Canada Nuclear Waste Management Organization contract to JCG/Dalhousie University. We are grateful to Clarence Pickett, Chris Hicks, and Kristin Henry (Vale Canada) for providing thick segments of the granophyre core near Sudbury, and for accompanying geological and spatial data. ChatGPT5.0 was used on Sep. 4 2025 to improve the curve fit and precise depth of the inflection point of synthetic TCN concentrations for Figure 3C, and those outputs were verified by including the synthetic sample data along with their curves, and by independently calculating where the energetic nucleon and slow muon production curves should meet for a give erosion rate.

# Financial support

CRISDal Lab is supported by CFI-MSI grant 45432 and NSERC-DG 06785-19. GR postdoc at Dalhousie was supported by the Early Postdoc Mobility Fellowship P2ZHP2\_199662, of the Swiss National Science Foundation (SNSF). LM's PhD is supported by NSERC-Doctoral scholarship and NS Graduate Scholarship at Dalhousie. MD's and SN's postdoc at Dalhousie were partially supported by the Canadian NWMO. LM Sudbury study was partially supported by the Canadian NWMO and training in AMS and <sup>36</sup>Cl target preparation at and by LLNL. Prepared in part by LLNL under Contract DE-AC52-07NA27344. This is LLNL-JRNL-2004624.

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
