# Peer review of "Muon paleotopometry"

_EGUsphere, 2025_

## Referee Comment (RC3)

Review of MS egusphere-2025-4370

Gosse et al. Muon paleotopometry

John Stone (stn@uw.edu)

1/ General comments

(i) Use of "μ" as an abbreviation for muon, as in μ-paleotopometry, μ-TCN, μ$_s$, μ$_f$, σ$_{μs}$, etc.  I found the frequent use of "μ" as a substitute for "muon" distracting.  It substitutes for a 4-letter word, so very little space is saved.  Various sections of the text can be condensed or eliminated (see below) to save much more space.  I also found it confusing:  I typically read the prefix "μ" as "micro", as in μs (microsecond), μg (microgram), etc, so almost every instance tripped me up.  I kept reading "micro-paleotopometry" and "micro-terrestrial cosmogenic nuclide", which is obviously not what the authors intended.

(ii) Length and organization: Several sections are longer than they need to be and some could be eliminated without harm to the purpose of the paper.  I suggest significantly reducing Section 2 (Background; lines 75-124) by referring to any of the dozens of good reviews or books in the cosmic ray literature which deal thoroughly with muon physics, muon production, muon interactions in air, water and rock, etc.  The relevant material has also been summarized in almost all previous cosmogenic nuclide papers on muon-produced nuclides underground.

I also suggest splitting off two new (shortened) sections starting at line 125.  The first discusses how cosmogenic nuclides are used to estimate erosion rates (lines 125-185).  The basic idea is very well established and could be greatly condensed.  The second part of this, which discusses "knees" in nuclide concentration profiles, could be condensed or perhaps even eliminated (I found it very confusing; see comments below).

In addition lines 189-293, which contrast the use of muons for measuring erosion rates/changes in surface cover/etc, could become a new section. This section discusses most of the really new and important ideas in the paper - how muon-produced nuclides at depth are sensitive to geomorphic changes over much larger areas and over much longer timescales than spallogenic nuclides, how the the muon flux at depth is insensitive to all the complications of latitude-scaling and paleomagnetic variation that plague spallogenic nuclides, etc.  This material deserves more prominence and clarity, which I think would result from condensing the previous 5-6 pages of review and placing it in a separate section.  The first sentence (line 189: "Why consider deep muogenic nuclides?") could even be converted to the section heading.

With regard to organization, I suggest moving the section on measurement precision to the end, or perhaps to an appendix.  Its current position seems to me to break up the flow from Section 3,

about the design of deep-sampling experiments, to Section 5.2, which describes examples of such experiments. [Note, there is no Section 4 currently].

Section 5.4, about muon transport codes and flux models, might also go in a separate section or an appendix. This is an important part of the paper. Models like MUSUN and MUTE are important for calculating the muon flux at depths where range fluctuations become important, and where awkward surface topography complicates analytical muon flux calculations. However it seems out of place between the Swiss Alps example (Section 5.3) and the inverse erosion-rate calculation (5.5).

2/ Specific comments

L.25 Last sentence of the abstract. As written, I don't understand this sentence. After eventually reading the material in Section 5.5, I think it is trying to say: "Cosmogenic nuclide concentrations are more sensitive to recent erosion than to erosion rates in the distant past". Also, if that's what the authors are trying to say, it's well known and probably doesn't need to be stated in the abstract.

L.49 and 54 Use the terms "mass depth" and "shielding thickness" for what I think is the same thing. I prefer the terms "depth" for conventional depths in centimeters or meters and "shielding depth" for the depth x density product in g cm$^{-2}$, but "mass depth" is so widely used in the cosmogenic nuclide literature that I'm pretty sure I'm out-voted. Note also that $S_i$ and $S_f$ are currently defined in the caption to Fig. 1.

L.71 "samples deeply in crust with ..." should be "samples deep in the crust with ...". Maybe it would be better to say "at depths up to a few hundred meters". Petrologists would say that "deep in the crust" means tens of kilometers.

S.2 Curiously there's no mention of the muon half-life, though L.87 notes than muons can decay.

L.83 Attenuation of the muon flux has nothing to do with muons being "smaller" than fast neutrons. Fast nucleons (hadrons) interact via the nuclear strong force, muons do not, so muons move through matter much farther than protons and neutrons of the same energy.

L.85 and following: Because this is mostly about how muons produce cosmogenic nuclides, it would be simpler to divide the discussion into "fast muons" and "stopped negative muons" rather than "fast" and "slow" muons. The production mechanisms are well described in Heisinger's two papers in 2002, and in many previous papers. This, and all of the Section 2 preamble about atmospheric and near-surface cosmogenic nuclide production, could be condensed by referring to a few relevant papers or Tibor Dunai's textbook.

L.100 Decametres and further on, hectometres. I am all in favour of these SI prefixes, but because they aren't nearly as common as "centi", "kilo", etc it might benefit some readers to

define them the first time each is used in the text e.g. "greater than decametres (tens of metres) ...". [Note - both show up in the abstract, but should probably not be defined there].

L.100 "... uncertainty in the energy spectra for $\mu_f$ over million-year timescales ...". Is there any? If so, how is this known? Are uncertainties due to conflicting data? I am not sure what could be measured to characterise muon energy spectra in the distant past. Please provide a reference or references, or omit this statement. It's also worth noting that depth profiles that have been used to calibrate cosmogenic $^{36}Cl$, $^{26}Al$ and $^{10}Be$ production by muons were generated over hundred-thousand- to million-year timescales. So these reflect some kind of average of the fast muon spectrum over their build-up histories.

L.104-107 Two comments: (i) The Beacon Heights $^{10}Be$ and $^{26}Al$ calibrations definitely do not reproduce Heisinger's muon production parameters. For $^{10}Be$, the mismatches in estimates of parameters $f^*$ and $\sigma_0$, for stopped negative muon production and fast muon production, are 0.27x – 0.46x and 0.44x – 0.79x respectively (the range for each parameter reflects the value of $\alpha$ used to calibrate the fast muon yield; $\alpha$ cannot be resolved from existing depth-profile calibrations). Why the parameters determined by Heisinger differ so dramatically from the depth-profile calibrations remains a mystery, as far as I know. (ii) In addition to Balco (2017) please also cite Borchers et al. (2016) Geological calibration of spallation production rates in the CRONUS-Earth project. *Quaternary Geochronology*, *31*, 188-198. This was the first study to derive muon production parameters from the Beacon Heights depth profiles using Heisinger's production model.

L.109 Should also mention MUSIC and MUSUN, a pair of codes for modeling the muon flux underground, used in lots of early muon tomography work. The citation is: Kudryavtsev, V. A. (2009). Muon simulation codes MUSIC and MUSUN for underground physics. *Computer Physics Communications*, *180*(3), 339-346.

L.128 Equation should probably be numbered. Should also note that this equation is an approximation that only applies to cases where erosion rates are high such that $\rho \varepsilon / \Lambda \gg \lambda$. i.e. the full equation $N = P / (\lambda + \rho \varepsilon / \Lambda)$ can be simplified by neglecting $\lambda$. The simplified version shown is relevant to long-lived spallogenic nuclides for erosion rates greater than a few meters per million years. But not necessarily for nuclides produced at great depths by muons. For the purposes of this paper, the qualification is very important, because $\Lambda_\mu$ for nuclide production by muons is much greater than $\Lambda_{sp}$ for near-surface production by spallation. At the depths dealt with in this paper, where production is entirely by muons, nuclide build-up is much less sensitive to erosion than at shallow depths. The table below shows approximate attenuation lengths for nuclide production at depths in the range ~ 40 m - 300 m. The third column shows the loss rate to erosion for $\varepsilon = 10$ m/Myr. At this erosion rate, radioactive decay is more important than erosion in determining nuclide concentrations at depths below ~ 150 m for $^{10}Be$ and below ~ 60 m for $^{26}Al$. i.e. at these depths, the simple equation is wrong by roughly a factor of two (and gets increasingly incorrect at greater depths). In practical terms, at these depths only half the nuclide

concentration is sensitive to the erosion history of the overlying surface.  Hence for example, for $^{10}$Be in quartz at 150 m beneath a surface eroding at 10 m/Myr, the steady-state concentration will be roughly 1150 atom/g, but a factor-of-two change in erosion rate only changes the concentration by ~ 150 atom/g.  This is about one quarter of the sensitivity implied by the equation currently in the text.  The fractional change for $^{26}$Al, with its larger decay constant, is even smaller.

| Depth | Depth* | $\Lambda$ (approx) | $\rho \, \varepsilon \, / \, \Lambda$ (yr$^{-1}$) | $(\rho \, \varepsilon \, / \, \Lambda) \, / \, \lambda_{Be-10}$ | $(\rho \, \varepsilon \, / \, \Lambda) \, / \, \lambda_{Al-26}$ |
|---|---|---|---|---|---|
| (m) | (g cm$^{-2}$) | (g cm$^{-2}$) | for $\varepsilon$ = 10 m/Myr | | |
| 40 | 10000 | 1900 | 1.37E-6 | 2.74 | 1.38 |
| 80 | 20000 | 3210 | 8.11E-7 | 1.62 | 0.82 |
| 160 | 40000 | 6110 | 4.26E-7 | 0.85 | 0.43 |
| 240 | 60000 | 9120 | 2.85E-7 | 0.57 | 0.29 |
| 320 | 80000 | 12000 | 2.17E-7 | 0.43 | 0.22 |

* for $\rho$ = 2.5 g cm$^{-3}$

L.130  $\Lambda$ is the attenuation length for the overall production process, not for the particles responsible.

L.140-149  The language in this section is very complicated.  It's simpler to observe that steady-state is reached after erosion of 2-3 $\Lambda$ (again, for the case where $\rho \, \varepsilon \, / \, \Lambda \gg \lambda$).  This is why the spallation-dominated top of a depth profile reaches erosional steady state much more rapidly than the deep muon-dominated part ($\Lambda_\mu \gg \Lambda_{sp}$), and why even single-nuclide depth profiles can be used to solve for both the age and erosion rate of a surface, or the recent and initial erosion rates of a surface with a complex erosion history.  I know reviewers are not supposed to refer authors to their own work, but this is covered in detail in section 5.2 of Stone et al. (1998) Cosmogenic chlorine-36 production in calcite by muons. GCA 62, 433-454.  See in particular Figs. 8 and 11.

L.150-185  Discussion of "knees" in depth profiles.  There are several problems with this section.

(i) There is no mathematical definition of the "knees" referred to.  The caption to Fig. 3 mentions a method of finding inflection points, but doesn't specify how the "knees" plotted in Fig. 3C were located.

(ii) The inflection point between spallogenic and muon-induced production would be much easier to see if the profiles in Figs. 3B and 3C were drawn in log(N) vs linear(z) co-ordinates as shown below.

[Figure]

[Figure]

(iii) I doubt there is a robust mathematical way to find or define an inflection point between parts of depth profiles where nuclide production by negative muon capture is significant and where fast muon reactions dominate. The difference in their depth profiles is slight, and in the cases of $^{10}$Be and $^{21}$Ne (shown above) production via stopped negative muon capture is minor ($^{10}$Be) or zero ($^{21}$Ne; see Balco et al. 2019).

(iv) Fundamentally there is less information in the gradient of a depth profile than in the concentrations (gradient is the derivative of the concentrations). Calculating gradients is inherently sensitive to concentration errors, which will propagate significantly into locating an inflection point from a pair of gradients.

(v) The position of the spallation to muon-dominated inflection point is probably more sensitive to altitude of the profile surface than to its erosion rate (I haven't calculated this, but the sensitivity is clear - spallogenic production roughly doubles with every kilometer altitude; muon production at 3-5 m depth varies by less than 5% per kilometer).

L.195-199 "... less than a hectare ..." Should this be less than ~1 m$^2$? Cosmic rays responsible for near-surface production pass through a narrow cone (~ 90% within a 60° cone around the zenith). A hectare is a circle with radius ~ 56 m. Even large objects (e.g. boulders) 56 m away don't significantly affect near-surface nuclide production. Effects are fractions of a percent.

L. 201 The same arguments carry over into this section, where it's argued that topography 10 km away has a significant effect on nuclide production rates deep underground. Again, I think this is mistaken. The flux of muons with range 10 km in rock is very close to zero. Even though the muon flux broadens with increasing energy, the increase in slant range for muons travelling at angles close to the horizon is so large (slant range increases as $1/\cos(\theta)$, where $\theta$ is zenith angle) that such muons contribute minimally to nuclide production. As shown in the diagram below,

[Figure]

[Figure]

Production contained within the cone angle θ
through erosion of one attenuation length

*Cumulative fraction of nuclide accumulation due to cosmic rays travelling at zenith angle θ as erosion removes one attenuation length Λ from the overlying surface (i.e. in build-up of ~ 63% of the eventual steady-state concentration). Note Λ ~ 7.5 m at 40 m depth, for r = 2.5 g cm$^{-3}$.*

production by muons travelling at zenith angles > 60-70 degrees is insignificant in nuclide build-up at depths up to 40 m. I haven't done the calculation for hundred-meter depths, but I doubt the result would change significantly. To reach 40 m, a muon travelling at 60° to the zenith requires energy > 48 GeV. At 100 m, the required energy is > 130 GeV. Fluxes at such energies are much smaller than the near-vertical flux which reaches the same depths. See also Fig. 4 of Heisinger (2002) on fast muons, which shows how the $\cos^n(\theta)$ distribution of muon arrivals underground narrows around the zenith at depths below 1000 hg cm$^{-2}$ (roughly 40 m depth in rock).

L. 225 [Figure 4]. See the discussion and figure above. It's not the surface area subtended by a zenith-angle cone that matters. The area has to be weighted by the fraction of nuclide production due to the muons that pass through it. It's neat that muon-produced nuclides at tens- to hundred-meter depths provide information about erosion over much larger areas than near-surface samples, but the underlying geometrical argument does not imply that there's information about erosion over thousands of square kilometers. Based on the same calculation as the figure above, the area contributing 90% of the production to a sample 40 m beneath an eroding surface is ~ 5000 m$^2$. This area is contained in the circle of radius ~ 40 m above the sample point.

I suggest omitting or replacing Fig. 4.

L. 251-259 Discussion of geomagnetic effects on muons reaching hundred-meter depths, and Table 1. There's a lot more here than is needed. The discussion could be summarized by saying that muon energies required to reach hundred-meter depths (> 60 GeV) are significantly greater than geomagnetic cut-offs (< 18 GV). The primary particles whose interactions produced the

muons had even higher energies. Hence there is no significant latitude effect on the muon flux reaching such depths.

Re Table 1. I'm not entirely sure, but I think columns 3 and 4 are referring only to vertical muon spectra. If you integrate over zenith angle $\theta$, the median energy of muons reaching the depths in column 1 would be even greater than the values in columns 3 and 4. This strengthens the argument.

L. 266-269. This seems like an unwise argument. If concentrations were really invariant (" ... serve as replicates ..."), they would be insensitive to erosion. What's being said is basically re-stating the discussion about the long attenuation length $\Lambda$ for muon-induced production, covered in lines 140-149 above. It would be easier to say that deriving an erosion rate or erosion history from a depth profile at tens- to hundreds of meters will be most sensitive if the samples are widely spaced down the profile.

L. 306 This also re-states discussion about cut-offs and the absence of latitude effects for deeply-penetrating muons (lines 250-259).

Fig. 7 (L. 370) Figure caption refers to "isoeroderes", which I think is a made-up word. The curves on the figure are depth profiles calculated for different erosion rates, which seems like a simpler description. Also, the figure and the final sentence of the caption could be used to replace a lot of the complicated text in the section on $\Lambda$ and its effect on sensitivity of the depth profile to surface erosion (lines 140-149; see my notes and table above). The statement that "The wider spacing [between profiles] toward the top provides the improved resolution of erosion rates." is again demonstrating that the profile is more sensitive to erosion where $\Lambda$ is shorter, and less sensitive deeper in the profile where $\Lambda$ is long.

The y-axis of Fig. 7 could be more clearly labeled. *e.g.* indicate units of $[10^4$ g cm$^{-2}]$ rather than the "x$10^4$" annotation at the top of the diagram.

L. 382 The limitation for $^3$He and $^{21}$Ne is almost always the difficulty of distinguishing small amounts of *cosmogenic* $^3$He and $^{21}$Ne from much larger amounts of nucleogenic $^3$He and $^{21}$Ne built up over the lifetime of the rock. The nucleogenic production rates are usually small, but in cases where the He and Ne closure ages are tens or hundreds of millions of years, build-up will likely swamp muon-induced build-up.

L. 400-404. See comment above. A similar limitation applies to $^{36}$Cl if the mineral analyzed contains significant $^{35}$Cl, resulting in $^{35}$Cl(n,$\gamma$)$^{36}$Cl production from radiogenic neutrons. Accurate estimation of ($\alpha$,n) neutron production is notoriously difficult. This is a significant source of uncertainty even for surface exposure dating of young, Cl-rich samples.

L. 426 "The mine stope runs laterally ...". I think the term should be "adit", "drift" or "tunnel". A stope refers to a vertical or cavernous opening. Adits, drifts and tunnels run horizontally.

L. 441 / Fig. 9    Consider adding a depth scale to the edge of Fig. B or an indication of the depth between the valley axis and the line of samples.  The necessary information is given in the bottom panel of Fig. 10, so this is not a big deal, but it would be useful in looking at Fig. 9B.

Fig. 10 Top panel and caption:  What is being plotted as the "Erosion Rate" in the top panel? From the correspondence between the top and middle panels it looks like what has been calculated is a the $2\pi$ steady-state surface erosion rate that would correspond to each of the measured $^{10}$Be concentrations.  If so, I don't think it's a very helpful measure to be plotting. Based on the model sketches and description, the samples won't be at steady-state and don't have $2\pi$ exposure geometry, and the surface above each sample has a 2-stage (or more complicated) erosion history.

L. 455-520  Discussion of the horizontal transect experiment.  Overall, this is a very neat experiment, but I got lost in the complicated discussion.  It would be good to examine more realistic solutions based on the combined data (all of which are sensitive to the erosion history of the valley) rather than going through the data one-by-one.  With a few geometrical simplifications (e.g. assuming a fixed valley profile and stream gradient), one could forward-model $^{10}$Be concentrations at each sample position for cases such as slow, steady valley incision over millions of years, steady valley incision starting at some time in the past, very rapid incision at some time in the past, with slow or zero erosion before and after, etc.  Going through the exercise should reveal whether any such geomorphic histories are more likely, which can be ruled in or out, whether data from surface samples would be useful, whether it would be helpful to obtain data from more samples underground, etc.  It would be a good illustration of the goal of the paper, which is to show how deep, muon-produced nuclide concentrations can provide geomorphic information over large areas and farther back in time than can be obtained from surface samples.  [Note - this is also a lot of work.  While it would be a great addition to the paper, getting it done shouldn't be a barrier to publication].

The word "isoerodere" is used again in line 484 when describing a depth profile.  See note re line 370.

Fig. 11.  Inexplicable depth-profile comparisons.  See two previous notes above - the data need to be considered as a whole, it makes no sense to consider them one-by-one.  Fig. 11 compares $^{10}$Be concentrations, which are very unlikely to be at steady state, to simple steady-state depth profiles.  I suggest omitting it.

L. 543.  "rock distances beyond 50 km"  This has to be a misprint.  As far as I know no experiment has ever observed a muon with a range of 50 km in rock.  Even if such energetic muons are occasionally produced, their flux (and contribution to nuclide production) would be negligible compared to muons arriving from close to the zenith.

Note also that topographic calculations out to 50 km distances need to take the curvature of the Earth into account. Topography 50 km away has to be taller than ~ 200 m before it's even visible over the horizon.

Fig. 13 caption Zenith angle range $\theta$ = 0-75°. Based on the slant range effects noted above (comments re lines 201-225) this should be adequate for almost all calculations. Only very specialized cases (e.g. involving steep slopes, depth profiles into narrow mesas, etc) are likely to need broader angular coverage.

L. 583-620 4-stage erosion calculation. (i) For the hypothetical case considered, the discussion should also mention that sensitivity to erosion in the first 2 stages (8-4 Ma and 4-2 Ma) is limited by the nuclide half-lives. 4 Myr is nearly 3 half-lives of $^{10}$Be and nearly 6 half-lives of $^{26}$Al, so fewer than 1/8 of the $^{10}$Be atoms produced in the first stage (and essentially none of the $^{26}$Al atoms) remain in the present-day sample. Al-26 will barely remember the second stage (4-2 Ma) either. Coupled with the fact that the hypothetical sample was 70 - 90 m deeper (with production rates correspondingly lower) during these initial stages in the geomorphic history, there's very little chance of recovering accurate erosion rates so far back in time. [Realistically, the amount of $^{10}$Be (and even more so, $^{26}$Al) surviving in the sample from > 4 Myr BP is likely to be smaller than the uncertainty on the concentration measurement].

L. 589 Simplifying the calculation to an exponential approximation with a fixed attenuation length of 5700 g cm$^{-2}$ is not valid. This is the approximate attenuation length for production at ~ 30 m depth, but the calculation involves much greater depths (50–300 m) where the attenuation length is much greater. At 300 m, it is ~ 25000 g cm$^{-2}$.

L. 611 "... searching for four erosion rates with two isotopes". The specific problems are: (i) the two isotopes have similar production profiles and attenuation lengths, so there is not very much sensitivity to depth in their accumulation rates. (ii) The $^{26}$Al half-life is too short to be useful for the first two stages in the erosion history. See note above.

L. 641 Should read "Beacon Heights site, Antarctica" and should cite Borchers et al., 2016 as well as Balco, 2017. See note re L. 104-107 above.

L. 657 Low-level $^{10}$Be carrier will be essential, but there's no indication that Al carrier needs to be from a deeply-shielded source. All commercial Al I've ever measured has $^{26}$Al below detection limits ($^{26}$Al/$^{27}$Al < n x 10$^{-16}$). In fairness, it is always worth confirming that Al carrier (which typically comes from bauxite-derived commercial Al) is free of $^{10}$Be. In the case of $^{36}$Cl, Weeks Island Halite, which is widely used as a Cl carrier, (i) is from a deeply-shielded (salt-dome) source, (ii) has a $^{36}$Cl/Cl ratio < 5 x 10$^{-17}$ (Fifield, L. K. et al. (2013). Ultra-sensitive measurements of $^{36}$Cl and $^{236}$U at the Australian National University. *Nucl. Instr. Meth. B*, *294*, 126-131).